# Disintegrin-like Protein Strategy to Inhibit Aggressive Triple-Negative Breast Cancer

**DOI:** 10.3390/ijms241512219

**Published:** 2023-07-30

**Authors:** Inès Limam, Mohamed Abdelkarim, Mohamed El Ayeb, Michel Crepin, Naziha Marrakchi, Mélanie Di Benedetto

**Affiliations:** 1Laboratory of Biomolecules, Venoms and Theranostic Applications, LR20IPT01, Institut Pasteur of Tunis, Tunis El Manar University, Tunis 1068, Tunisia; 2INSERM Unité 553, Laboratoire d’Hémostase, Endothélium et Angiogenèse, Hôpital Saint-Louis, 75010 Paris, France; mohamed.abdelkarim@fmt.utm.tn (M.A.);; 3LR99ES10, Faculty of Medicine of Tunis, Tunis El Manar University, 1 Rue Djebal Lakhdar, Tunis 1006, Tunisia; 4IUT of Saint-Denis, Department HSE, Université Paris 13, UMRS941 SMBH, 1 Rue de Chablis, 93000 Bobigny, France

**Keywords:** disintegrin-like, breast cancer, TNBC, HUVEC, migration, angiogenesis

## Abstract

Venoms are a rich source of bioactive compounds, and among them is leberagin-C (Leb-C), a disintegrin-like protein derived from the venom of *Macrovipera lebetina transmediterrannea* snakes. Leb-C has shown promising inhibitory effects on platelet aggregation. Previous studies have demonstrated that this SECD protein specifically targets α5β1, αvβ3, and αvβ6 integrins through a mimic mechanism of RGD disintegrins. In our current study, we focused on exploring the potential effects of Leb-C on metastatic breast cancer. Our findings revealed that Leb-C disrupted the adhesion, migration, and invasion capabilities of MDA-MB-231 breast cancer cells and its highly metastatic D3H2LN sub-population. Additionally, we observed significant suppression of adhesion, migration, and invasion of human umbilical vein endothelial cells (HUVECs). Furthermore, Leb-C demonstrated a strong inhibitory effect on fibroblast-growth-factor-2-induced proliferation of HUVEC. We conducted in vivo experiments using nude mice and found that treatment with 2 µM of Leb-C resulted in a remarkable 73% reduction in D3H2LN xenograft tumor size. Additionally, quantification of intratumor microvessels revealed a 50% reduction in tumor angiogenesis in xenograft after 21 days of twice-weekly treatment with 2 µM of Leb-C. Collectively, these findings suggest the potential utility of this disintegrin-like protein for inhibiting aggressive and resistant metastatic breast cancer.

## 1. Introduction

Angiogenesis is a crucial process involved in various physiological and pathological events, including embryonic development, tumor growth, metastasis, and inflammatory disorders [1,2]. Tumors rely on angiogenesis to obtain oxygen and nutrients, enabling invasion and metastasis [3,4]. The process of angiogenesis encompasses several steps, such as breakdown of the basement membrane, migration and proliferation of endothelial cells, formation of lumens, connection of new blood vessels with existing circulation, and extensive remodeling of the extracellular matrix (ECM) [5,6]. Integrins, a family of multifunctional cell adhesion receptors, play a crucial role in angiogenesis by anchoring cells to the ECM and serving as a vital link to the cytoskeleton, enabling stable cell adhesion, growth, and migration [7,8,9]. Various integrins have been implicated in tumor angiogenesis, including α2β1, α5β1, αvβ5, and αvβ3, with studies highlighting their potential to suppress neovascularization and tumor progression [10,11,12]. Consequently, exploring these integrins as novel anticancer strategies has gained significant attention [13,14]. Research has demonstrated that αvβ3 collaborates with growth factor receptors such as vascular endothelial growth factor (VEGF) and basic fibroblast growth factor (FGF2) to promote angiogenic events, primarily mediated by the activation of matrix metalloproteinases (MMPs) and proteolytic degradation of the ECM, facilitating proliferation and migration of endothelial cells to the tumor site [8,11,15]. It should be noted that targeting tumor-induced angiogenesis has traditionally centered on inhibiting the VEGF signaling pathway. However, a major challenge associated with current VEGF-based anti-angiogenic treatments is the development of resistance. This resistance often arises from the upregulation and compensatory mechanisms of other growth factors, with FGF2 being one of the most prominent factors involved in this process [16]. Due to its strong stimulation by FGF-2, αvβ3 integrin presents a promising approach for overcoming resistance to anticancer drugs. Furthermore, this integrin has also demonstrated a prosurvival function: upon binding to fibronectin, endothelial cells are protected from apoptosis through the activation of various signaling cascades [17]. In the context of breast cancer, the overexpression of αvβ3 integrin has been observed to be associated with the development of bone metastasis and is known to facilitate increased tumor growth and invasion in response to osteopontin. Additionally, αvβ3 expression plays a significant role in regulating the response of breast cancer cells to chemotherapy, thereby serving as a potential marker for chemosensitivity. It has been noted that αvβ3 integrin is upregulated in cells treated with microtubule interfering agents, while its expression remains unaffected in cell lines that display resistance to these drugs. Notably, the enforced expression of the β3 subunit has been shown to confer increased resistance to paclitaxel in cancer cells [17,18,19]. Another critical marker of induced angiogenesis, α5β1 integrin, is associated with tumor malignancy, invasiveness, and metastasis formation in response to various stimuli, such as FGF2 [20]. In breast cancer, elevated expression of α5β1 is linked to increased invasive capacity and regulates cell invasion by modulating MMP2 levels. It also plays a role in doxorubicin resistance by enhancing the activity of AKT, mTOR, and ERK1/2 protein kinases [21]. Tumor cells exhibit distinct integrin patterns depending on their specific requirements and stage of development. Therefore, targeting both cancer cells and endothelial cells at the tumor level could provide a more effective strategy for cancer treatment and overcome resistance, as opposed to solely focusing on individual cancer cells [14,22,23,24]. Indeed, numerous studies on potential drugs derived from natural sources have demonstrated that natural molecules can serve as a valuable source of novel drugs due to their unique chemical characteristics and biological activities [25,26]. 

Certain natural toxins derived from snake venom possess specific and potent inhibitory effects on integrin functions [14,27]. Interestingly, it has been reported that the sensitivity of endothelial cells to external beam radiotherapy is enhanced when administered concurrently with an integrin antagonist in an animal model of prostate cancer [28]. Among these toxins, disintegrins are non-enzymatic proteins that are released from precursor forms. Snake venom metalloproteases (SVMPs) are zinc-dependent proteinases classified into three classes (I-III) and several sub-classes based on their domain composition [29]. Class III SVMPs consist of a metalloproteinase domain (M), a disintegrin-like domain (D), and a cysteine-rich domain (C). Some class III proteins undergo autoproteolytic cleavage, resulting in a stable protein composed of the D and C domains [30]. This processed D/C domain, also known as a disintegrin-like protein, has been found to possess anti-adhesive properties [27]. Many of these naturally occurring or recombinant proteins act as potent antagonists of the α2β1 integrin collagen receptor [31,32]. Leberagin-C (Leb-C), a disintegrin-like protein containing the SECD motif instead of the RGD recognition motif, was isolated from *Macrovipera lebetina transmediterrannea* snake venom. It has been shown to be a potent inhibitor of platelet aggregation induced by thrombin and arachidonic acid [33]. Studies have demonstrated that Leb-C specifically inhibits α5β1, αvβ3, and αvβ6 integrins through a mechanism mimicking RGD disintegrins derived from class P-II SVMPs [33]. 

In our current study, we investigated the effects of Leb-C on various steps of angiogenesis by examining its impact on the adhesion, migration, invasion, and proliferation of human umbilical vein endothelial cells (HUVECs) in vitro. Additionally, we conducted evaluations of the anti-tumor effects of Leb-C on highly metastatic breast cancer cells, MDA-MB-231 and its aggressive subclone D3H2LN, in vitro and in vivo using nude mice as an experimental model.

## 2. Results

### 2.1. Leb-C Inhibited MDA-MB-231 and D3H2LN Cell Adhesion

Multiple integrins are expressed in cancer cells, enabling their adhesion to extracellular matrices. We have previously reported that Leb-C inhibits tumor cell adhesion to various substrates by disrupting cell/ECM interactions through an integrin-dependent mechanism [33]. To investigate the effect of Leb-C on breast cancer, we initially conducted a cell adhesion assay using MDA-MB-231 and D3H2LN cell lines. The cells were pre-incubated with increasing concentration of Leb-C (0–1.2 µM) for 30 min before being allowed to adhere to immobilized fibronectin (Fn). As depicted in Figure 1A, Leb-C significantly inhibited the adhesion of both cell lines to Fn. This effect was dose-dependent, with half inhibitory concentration (IC50) values of 0.45 µM and 0.55 µM for MDA-MB-231 and D3H2LN, respectively (Figure 1A).

### 2.2. Leb-C Suppressed MDA-MB-231 and D3H2LN Migration and Invasiveness

As migration is a fundamental characteristic of malignant tumor cells, we explored Leb-C effects on the migration of breast cancer cell lines using a transwell assay. MDA-MB-231 cells and the D3H2LN clone population typically migrate to the lower side of the chamber when 10% foetal bovine serum (FBS) is used as a chemoattractant. It is worth noting that after 16 h of treatment, Leb-C demonstrated a highly dose-dependent effect on the migration of both cell types (Figure 1B). The maximum dose effect (100%) was observed at the dose of 0.5 µM, and the IC50 values were approximately 0.01 µM and 0.025 µM for MDA-MB-231 and D3H2LN cell lines, respectively (Figure 1B). 

Since invasion is also a crucial process in metastasis, we investigated the invasion of MDA-MB-231 and D3H2LN cells. In these invasion assays, breast cancer cells invaded inserts precoated with Matrigel Matrix. Figure 1C reported that 24 h of exposure to 1 µM Leb-C significantly reduced the invasion of MDA-MB-231 and D3H2LN cells by 80% and 65%, respectively. The IC50 for this inhibition was about 0.1 µM for both cell types (Figure 1C).

To examine whether the effects of Leb-C were due to its growth-suppressing activity, a cell proliferation assay was carried out. It should be noted that up to a concentration of 2 µM, Leb-C did not affect the proliferation of MDA-MB-231 and D3H2LN cells after three days of exposure. Thus, Leb-C dose-dependently inhibited breast cancer cell adhesion, migration, and invasion (*p* < 0.05). 

To investigate the role of Leb-C on angiogenesis, which are key players in the process of tumor progression and treatment response, we used human umbilical vein endothelial cells (HUVECs).

### 2.3. Leb-C Inhibited HUVEC Adhesion 

Generally, HUVECs adhere to immobilized Fn. Therefore, we evaluated the impact of increasing concentrations of Leb-C on the adhesion of HUVECs to Fn. As illustrated in Figure 2A, Leb-C exhibited dose-dependent inhibition of HUVEC adhesion to Fn after 30 min of pre-incubation, with an IC50 of 0.58 µM.

### 2.4. Leb-C Suppressed HUVEC Cell Growth

We also investigated the impact of Leb-C on angiogenesis using an HUVEC proliferation assay system stimulated by the proangiogenic factor FGF2, without the presence of an extracellular matrix (ECM) to eliminate any potential anti-adhesive effects of the protein. Our findings, as shown in Figure 2C, demonstrated that Leb-C had a significant influence on HUVEC cell growth, leading to cellular rounding and loss of intercellular contact. Specifically, Leb-C effectively inhibited HUVEC proliferation in a dose-dependent manner after 72 h of treatment (Figure 2B). The maximum observed effect, with an inhibition rate of 85%, was achieved at a Leb-C concentration of 2.3 µM, and the IC50 value was approximately 1.17 µM. These results provide compelling evidence of the potent anti-proliferative activity of Leb-C against HUVECs, indicating its potential as a promising inhibitor of angiogenesis.

### 2.5. Leb-C Exhibited a Reduction in HUVEC Migration and Invasion

Angiogenesis relies heavily on the motility and invasion of endothelial cells. Thus, we evaluated the effect of Leb-C on HUVEC migration and invasion towards FBS and FGF2. A migration assay was performed using transwell inserts coated with 0.25% gelatin, while invasion assays were conducted using Matrigel. Different concentrations of Leb-C were used, and the evaluations were carried out after 16 h for migration and 24 h for invasion. As depicted in Figure 3A, treatment with 0.5 µM of Leb-C resulted in a complete inhibition of HUVEC cell migration (*p* < 0.01). Likewise, 1 µM of Leb-C significantly suppressed HUVEC cell invasion by 85% (Figure 3B). Specifically, the IC50 values for these dose-dependent inhibitions were approximately 0.08 µM and 0.1 µM for migration and invasion, respectively (Figure 3). These findings demonstrate the potent inhibitory effects of Leb-C on both cell migration and invasion in HUVECs.

### 2.6. Leb-C Decreased D3H2LN Tumor In Vivo

For the in vivo experiment, we employed the D3H2LN tumor model in nude mice. Tumor mass was assessed 10 days after cell inoculation. The impact of Leb-C on tumor growth was evaluated using two approaches: mean measured tumor volume (mm^3^) and mean photon flux (photon/s/cm^2^) obtained through bioluminescence imaging of D3H2LN tumors using IVIS. Interestingly, when considering mice that developed tumors in each group, we observed that Leb-C led to an approximately 50% reduction in D3H2LN tumor uptake compared to the control after 25 days of treatment (Figure 4A). Furthermore, Figure 4B demonstrates that administration of 2 μM Leb-C, twice weekly for 21 days, significantly suppressed D3H2LN tumor volume by 73% (*p* < 0.01). Specifically, on the 25th day of treatment, the mean tumor volume in the treated group (approximately 177 mm^3^) was approximately 3.5 times smaller than that of the control group (approximately 653 mm^3^) (*p* < 0.01). Additionally, analysis of the bioluminescence images generated by the In Vivo Imaging System (IVIS) (Figure 4C) revealed significant results only after 21 and 24 days of twice-weekly treatment with 2 μM Leb-C, with an approximate 69% inhibition in the luminescence signal (reflecting D3H2LN cell density) compared to the control (*p* < 0.05). 

### 2.7. Leb-C Reduced Tumor Angiogenesis In Vivo 

To assess intratumoral angiogenesis in paraffin sections, endothelial cells were identified using GSL-1, a lectin that specifically binds to galactosyl residues present on the vascular endothelium. Histological analysis of the tumor at the end of the experiment revealed the presence of necrotic areas in the central regions of most tumors in the control group. Additionally, the analysis revealed a reduction in the number of blood vessels within viable tumor fields in the Leb-C-treated group compared to the control group (Figure 5). Notably, the vessel diameters were larger in the control group compared to the Leb-C-treated group (Figure 5A). Quantification of vessel density demonstrated that treatment with 2 μM Leb-C, administered twice a week, led to a significant 50% reduction in vessel density within viable tumor fields compared to the control group (*p* < 0.05) (Figure 5B).

## 3. Discussion

Breast cancer is a major cause of cancer-related death in women worldwide [34]. Despite progress in early detection and awareness, challenges such as drug resistance, recurrence, and metastasis persist, affecting overall survival rates [35]. Approximately 15–20% of breast cancer cases belong to the triple-negative breast cancers subtype (TNBCs), which is known to be the most aggressive subtype characterized by the absence of estrogen receptor, progesterone receptor, and human epidermal growth factor receptor 2 amplification [36,37]. The triple-negative phenotype is associated with higher rates of relapse, metastasis, and mortality compared to other subtypes of breast cancer [38]. Heterogeneity in breast cancer is not limited to the characteristics of tumor cells but also extends to the tumor microenvironment, influencing tumor progression and treatment response. The diverse cellular components, including stromal cells and extracellular matrix, interact with tumor cells and create a complex environment that can impact tumor behavior and treatment outcomes [35]. Metastasis involves integrin-mediated processes such as cell migration, invasion, and attachment regulation [20]. 

In our previous study, we demonstrated that Leb-C, a non-enzymatic disintegrin-like protein derived from class P-III SVMPs, influenced cell adhesion by interfering with the function of αvβ3, αvβ6, and α5β1 integrins [33]. In the present study, we investigated the effects of Leb-C on breast cancer using MDA-MB-231 cells (a TNBC cell line [39]) and D3H2LN cells (an aggressive subclone of MDA_MB-231 that enhanced tumor growth, angiogenesis, and metastasis in mice) [40]. Our results demonstrate that Leb-C significantly inhibited the adhesion of MDA-MB-231 and D3H2LN cells to fibronectin in a dose-dependent manner. Furthermore, Leb-C effectively reduced the migration and invasion of both cell lines (*p* < 0.01). However, Leb-C did not affect the proliferation of these metastatic breast cancer cells. Our findings suggest that Leb-C suppresses breast cancer adhesion, migration, and invasion through its interaction with αvβ3, αvβ6, and α5β1 integrins, employing an RGD mimetic mechanism as described in our previous studies [33]. Our data also emphasize the important role of these integrins in breast cancer invasion. αvβ3, in particular, is detected at the invasive front and distant metastases, and numerous studies suggest that αvβ3 expression enables tumor cells to invade and survive in hostile environments, indicating its role in the aggressiveness and metastatic potential of these tumors [12,20,41]. These studies suggest that inhibiting αvβ3 function in breast cancers may offer a strategy to suppress metastatic tumor cells [41]. Similar observations have been made for αvβ6 and α5β1, which are overexpressed in breast cancer cells and proposed as markers of tumor cell invasiveness [20,42,43]. To our knowledge, only one study has reported on the effects of the disintegrin-like protein family in breast carcinoma. Alternagin-C, an ECD-disintegrin-like protein, was identified as a potent antagonist of α2β1 integrin, capable of attenuating the adhesion of MDA-MB-231 cells to collagen I but not inhibiting cell migration [44]. However, there are limited studies describing the anti-tumor effects of the disintegrin-like protein family. Acurhagin-C, an ECD disintegrin from *Agkistrodon acutus* venom, has been reported to dose-dependently affect the adhesion of B16-F10 melanoma cells to immobilized fibronectin (IC50 of 0.65 μM) and disrupt their transendothelial migration [45]. Considering the high efficacy of Leb-C at low doses compared to other disintegrin-like proteins, it holds promise as a potent therapeutic agent for cancer treatment.

Breast cancer is known to be associated with the expression of integrins on the surface of tumor-associated vessels, which play a crucial role in angiogenesis signaling pathways [46,47,48]. Considering this, we examined the effect of Leb-C on endothelial cells using various in vitro assays, including cell adhesion, migration, invasion, and proliferation, using HUVEC as the model system. Our results revealed a significant dose-dependent inhibition of HUVEC adhesion to Fn in the presence of Leb-C. Moreover, Leb-C exhibited remarkable reductions in HUVEC migration and invasion induced by FGF2, following 16 h and 24 h incubations, respectively. Furthermore, exposure to Leb-C for 72 h suppressed FGF2-induced proliferation of HUVEC cells, with an IC50 value of approximately 1.17 µM. This finding is particularly significant considering that many tumors develop resistance to anti-VEGF treatments targeting tumor angiogenesis by promoting FGF2-dependent vasculature [49]. Additionally, it is known that FGF2-induced tumor angiogenesis is primarily mediated by αvβ3 and α5β1 integrins [8,20]. Leb-C likely exerts its action on HUVEC by interacting with these integrins, as previously described [33]. However, further mechanistic investigations are necessary to confirm these hypotheses. Nevertheless, our result is consistent with the previous findings on the disintegrin-like protein acurhagin-C, which inhibited FGF2-induced HUVEC proliferation by 50% at 400 µM, as well as adhesion to Fn (IC50 of 600 µM), migration, and invasion of these cells (IC50 of 200 µM and 100 µM, respectively) [50]. Furthermore, several reports on disintegrin-like proteins have highlighted their capacity to interfere with endothelial cell functions. For instance, leucurogin, a D/C protein cloned from *Bothrops leucurus* venom, inhibited HUVEC tube formation at 4.8 µM after an 18 h incubation period [51]. Another study demonstrated that 1 µM of alternagin-C, an isolated disintegrin-like protein from *Bothrops alternatus* venom, reduced various VEGF-induced angiogenic events in HUVEC, including adhesion (by 63%), viability (by 27%), migration (by 55%), and tube formation (by 53%) time [52]. Based on these studies, it appears that Leb-C exhibits greater efficacy in inhibiting angiogenic events compared to other disintegrin-like proteins. This highlights its potential as a potent inhibitor of angiogenesis, a critical factor in tumor growth, invasion, metastasis, and chemosensitivity. The significance of the endothelium within the tumor microenvironment cannot be understated, as it plays a pivotal role in these processes [35]. This is particularly significant as long-term treatment with anti-VEGF therapies can be toxic to some patients and may also induce resistance [53]. Moreover, recent research has revealed that the conditioned medium from cisplatin-treated endothelial cells promotes vasculogenic mimicry, which contributes to drug resistance and metastasis [54]. These findings collectively emphasize the promising potential of Leb-C as a highly efficient inhibitor of angiogenesis, providing valuable insights for its therapeutic application in combating tumor angiogenesis, metastasis, and resistance. 

The results obtained from in vitro analyses were further supported by an in vivo xenograft model using the highly metastatic and bioluminescent D3H2LN cell line. This model provides a sensitive system for evaluating breast cancer growth, dissemination, and response to anticancer therapies through bioluminescent imaging [40]. The experiments revealed that Leb-C led to an approximately 50% reduction in D3H2LN tumor uptake compared to the control group. Notably, the anti-tumor effects of Leb-C on D3H2LN xenografts were remarkable. Subcutaneous injection of 2 µM of Leb-C per mouse twice a week for 21 days significantly reduced the volume of D3H2LN tumors by approximately 70%. Additionally, the histological analysis of tumors showed that the blood vessels in the Leb-C-treated group had smaller calibers compared to the control group. Furthermore, Leb-C treatment resulted in a 50% decrease in vessel density in the treated group compared to the control group. To the best of our knowledge, Leb-C is the first disintegrin-like protein to demonstrate anti-tumoral and anti-angiogenic activity in vivo in a breast cancer model. For the first time, we demonstrated that a D/C protein could inhibit both angiogenesis and tumor uptake or proliferation in nude mice. This breakthrough discovery highlights the potential of Leb-C as a promising therapeutic agent for the treatment of metastasis. It is worth mentioning that the breast cancer tumor model inhibited by Leb-C, known as D3H2LN, represents a sub-population of cells characterized by high metastatic capacities and resistance to chemotherapy. This model specifically represents TNBCs, which are recognized as the most aggressive forms of the disease with a higher mortality rate attributed to a disproportionate number of metastatic cases [55]. Compared to other subtypes of breast cancer, TNBC exhibits greater resistance to conventional treatments such as surgery, radiation therapy, and neoadjuvant chemotherapy. The resistance observed in TNBC may be linked to its highly invasive phenotype, EMT transition, resulting in decreased susceptibility to standard treatment agents and ultimately leading to a poorer prognosis for patients [55,56]. Despite advancements in breast cancer research, the underlying mechanisms driving the proliferation and metastasis of TNBC cells are still not fully understood. This lack of understanding contributes to the absence of a standard treatment regimen for these types of cancers, further exacerbating the overall poor prognosis [56]. Consequently, there is a critical need for innovative therapeutic approaches to address the limited treatment options available for TNBC. Therefore, the development of molecules such as Leb-C holds great interest.

Leb-C is the first disintegrin-like protein described to possess both in vitro and in vivo anti-angiogenic activities. This remarkable progress opens new possibilities in the field. Nevertheless, it is important to acknowledge the limitations of this study. The use of a primary cell culture has posed challenges in exploring the intricate mechanisms at play. Therefore, additional research is necessary to gain a comprehensive understanding of the underlying mechanisms, optimize administration protocols, and evaluate the potential synergistic effects of Leb-C in combination with existing therapies. Overall, Leb-C holds great promise as a lead compound for future investigations and development in the field of chemosensitization.

## 4. Materials and Methods

### 4.1. Compound Isolation

Venom was collected from *Macrovipera lebetina transmeditarannea* snakes at the serpentarium of Tunis Pasteur Institute and subjected to a purification process. The venom was loaded onto a Sephadex G-75 column (Pharmacia, Uppsala, Sweden). Subsequently, Leb-C was isolated following a previously described procedure [26]. In brief, Fraction I, obtained from the Sephadex G-75 column, was applied to reverse-phase high-performance liquid chromatography using a C8 column (250 × 4.6 mm, 5 mm; Beckman coulter, Marseille, France) that was pre-equilibrated with a solution of 0.1% trifluoroacetic acid in 20% acetonitrile. The elution was performed at a flow rate of 0.8 mL/min, employing an acetonitrile linear gradient of 20–65% over a period of 40 min.

### 4.2. Cell Lines and Cell Culture

The estrogen-independent breast human carcinoma, cell line MDA-MB-231 (American Type Culture Collection: ATCC HTB-26), was routinely grown in Dulbecco’s Modified Eagle’s Medium (DMEM) (Invitrogen, Waltham, CA, USA) supplemented with 10% FBS, 1% penicillin-streptomycin 10,000 U/mL (Invitrogen, CA, USA), and 1% Glutamine 200 mM (Invitrogen, CA, USA). The cells were maintained at 37 °C in an incubator (ASSAB T304 GF) with 5% CO_2_ and 95% air. D3H2LN cell line isolated from lymph node metastasis of MDA-MB-231 was obtained from Caliper Life Sciences (Alameda, CA, USA). D3H2LN is a metastatic subclone selected from a stable clone of MDA-MB-231 cells expressing firefly luciferase [57]. D3H2LN cells were cultured in Minimum Essential Medium with Earl’s Balanced Salts Solution (MEM/EBSS) supplemented with 10% foetal bovine serum, 1% nonessential amino acids, 1% L-glutamine, and 1% sodium pyruvate and antibiotics (all from Hyclone, Logan, UT, USA), at 37 °C in a humidified atmosphere containing 5% carbon dioxide. HUVECs were isolated and cultured following the methods described by Jaffe et al. [58]. Briefly, blood was flushed from human umbilical cord veins using cord buffer (0.14 M NaCl, 0.004 M KCl, 0.001 M phosphate buffer, pH 7.4, 0.011 M glucose). The veins were then filled with a 0.2% type I collagenase solution (Sigma, St. Louis, MO, USA) and incubated for 10 min at 37 °C [59]. The cord vein was flushed with M199 medium to collect detached cells, followed by centrifugation at 1200 rpm for 5 min. The endothelial cells were cultured in M199 medium containing 20% FBS, penicillin (100 U/mL), and streptomycin (100 mg/mL) for 18–24 h. The cells were subsequently washed during a medium change. HUVECs used in the experiments were between the second and fourth passages. These endothelial cells were identified by positive immunofluorescent staining for von Willebrand factor antigen.

### 4.3. Cell Adhesion Assay

Adhesion studies were conducted using 96-well plates that were pre-coated with adhesion molecules such as Fn [33]. HUVECs, MDA-MB-231 or D3H2LN cells (5 × 10^4^ cells per well), detached from culture dishes using 0.53 mM EDTA, were suspended in PBS containing 0.9 mM calcium and 0.5 mM magnesium. The cell suspensions were then centrifuged at 1200 rpm for 5 min and resuspended in PBS. Cells were pre-incubated with increasing concentrations of Leb-C (ranging from 0 to 1.5 µM) for 30 min and then seeded onto the coated wells for 1 h at 37 °C. After the incubation period, the wells were washed, and the adhered cells were fixed with 20% methanol (50 µL/well) for 30 min. Subsequently, the cells were stained with 0.5% crystal violet in methanol (50 µL/well) for 1 h at room temperature. Following staining, the wells were washed and incubated for 30 min with a solution of ethanol: citrate (0.1 M) in a 1:1 ratio at pH 4.2. Finally, the absorbance of the wells was measured at 540 nm using an ELISA reader (Labsystems, Kennett Square, PA, USA).

### 4.4. Cell Migration and Invasion Assays

Cell migration experiments were performed using migration chambers (Boyden chambers) with 8 µm pore size (Becton Dickinson, Bedford, MA, USA) following the previously described method [60]. The transwell permeable supports were hydrated with FBS-free medium for 1 h at room temperature. MDA-MB-231, D3H2LN, and HUVECs cells (2 × 10^5^ cells/mL) were resuspended in serum-free medium and preincubated with increasing concentrations of Leb-C (0, 0.01, 0.05, 0.1, 0.25, and 0.5 µM) for 30 min. The cell suspension was then carefully added to the upper chambers (insert). For MDA-MB-231 and D3H2LN cells, the lower chamber was filled with DMEM medium containing 10% FBS to serve as a chemoattractant. For HUVECs, the lower chamber was filled with M199 medium supplemented with 10% FBS and FGF2 (30 ng/mL) to promote chemotaxis. The Boyden chamber was incubated at 37 °C with 5% CO_2_ for 16 h. After incubation, the filter was gently removed from the chamber, and the cells on the upper side of the filter were removed by wiping with a cotton swab. The cells that had migrated to the lower surface of the filter, having penetrated the polycarbonate membrane pores, were fixed with methanol and stained with crystal violet solution (Sigma, St. Louis, MO, USA). Cell counts were performed in five randomly selected microscopic fields (×200 magnification) on the lower slide. The results were expressed as a percentage relative to the control, which was set as 100%.

To assess cell invasion through the extracellular matrix, invasion assays were carried out using Boyden chambers with 8 µm pore filters coated with Matrigel Matrix, a basement membrane protein mixture (Falcon, Corning, Acton, MA, USA) [60]. Cells were pretreated with Leb-C at concentrations of 0.1, 0.25, 0.5, and 1 µM before being transferred to the upper chamber. The chambers were then incubated for 24 h. The experiments were performed in triplicate, following the same protocol as described for the migration assay.

### 4.5. Cell Proliferation Assay

To evaluate the impact of Leb-C on cell proliferation, the MTT tetrazolium assay was assessed. HUVECs and MDA-MB-231 cells were seeded in 12-well plates at a density of 5 × 10^4^ cells per well in their respective culture medium and allowed to adhere for 24 h at 37 °C [61,62]. Subsequently, to stimulate HUVEC proliferation, FGF2 (30 ng/mL) was added to each well, followed by the addition of increasing concentrations of Leb-C (ranging from 0 to 2.5 µM). The cells were further incubated for an additional 48 and 72 h. After each incubation period, cell growth was determined by the MTT salt, which was converted to formazan by mitochondrial enzyme in active cells. Optical density was measured at 570 nm using a Labsystems Multiskan MS microplate reader. The experiments were performed in triplicate, with each Leb-C concentration tested in three wells.

### 4.6. Xenografts in Nude Mice

All in vivo experiments were conducted in compliance with the ethical guidelines of the French ethical committee and in accordance with the UKCCCR guidelines, as previously described [60]. The animals were housed in a temperature-controlled room with a 12:12 light–dark cycle and provided with ad libitum access to food and water. Four-week-old athymic nude mice (nu/nu, Janvier, France, n = 14) were randomly divided into a control group (n = 7) and a Leb-C-treated group (n = 7). The mice were subcutaneously injected with D3H2LN cells (2 × 10^6^ cells) suspended in 0.1 mL of PBS into the dorsal midline. In the treated group, Leb-C (2 µM) was included in the injection, while the control group received only PBS. One week after cell inoculation, the treatment group received 0.1 mL of Leb-C (2 µM) near the tumor, while the control group received 0.1 mL of PBS. This administration was repeated twice a week for 4 weeks. Tumor volume (mm^3^) was measured twice weekly using calipers to measure the major axes, and it was calculated using the formula V = 4/3π·R1^2^·R2, where R1 < R2.

### 4.7. Bioluminescent Imaging

Bioluminescence images were obtained from the dorsal view using the In Vivo Imaging System (IVIS) (Imaging System-50, Xenogen Corporation, Caliper Life Sciences, USA). The images were acquired approximately 5 min after intraperitoneal injection of D-luciferin to anesthetized mice [60]. To prevent saturation, the acquisition times were adjusted based on the signal strength. The bioluminescence signal was analyzed once a week using Living Image software (Xenogen Corporation, Caliper Life Sciences, USA). A region of interest (ROI) was drawn around the bioluminescence signal, and the photon flux (photon/s/cm^2^) was measured within the ROI.

### 4.8. Immunohistochemical Analysis

At the conclusion of the in vivo experiment, tumor specimens were collected, fixed in 4% paraformaldehyde, and embedded in paraffin using standard procedures. Serial sections of 5 µm thickness were routinely stained with hematoxylin and eosin. For immunohistochemical analyses, sections were deparaffinized and rehydrated. Endogenous peroxidase was inactivated using 3% H_2_O_2_ and then washed with TBS (Tris 0.05 M, NaCl 1.5 M, pH 7.6). Subsequently, sections were incubated with 10% normal goat serum for 1 h at room temperature to block non-specific binding. Endothelial cells were specifically labelled with GSL-1 isolectin B4 (Vector Laboratories, Burlingame, CA, USA) as previously described [60]. Briefly, sections were incubated with a 1:50 dilution of GSL-1 isolectin for 1 h at room temperature, followed by incubation with a goat anti-GSL-1 isolectin B4 antibody (1:400 dilution, Vector Laboratories) for 30 min. This was followed by incubation with biotinylated rabbit anti-goat immunoglobulins (1:400 dilution, Dako, Glostrup, Denmark) for 20 min in a humid chamber. After washing with TBS, the sections were treated with streptavidin-biotin peroxidase (LSAB kit, Dako) for 10 min and developed with diaminobenzidine tetrahydrochloride as the chromogenic substrate. Finally, the sections were washed in water and counterstained with hematoxylin.

Intratumoral microvessel areas were determined as described previously [60]. Ten non-sequential sections from each tumor were randomly selected for analysis. For both control and Leb-C-treated GSL-1-stained tumor sections, ten fields containing viable tumor cells, as indicated by hematoxylin staining, were randomly chosen for analysis. Using a Reichter-Jung microscope (Polivar, Vienna, Austria) at 100× magnification, each tumor section was scanned to identify and select the areas with the highest vascular density based on criteria established by Weidner et al. [63]. Two or three photographs were taken at a 250× zoom level for each section. Each count represented the highest number of microvessels identified in any 250× field (1.02 mm^2^). Individual vessels were identified as GSL-1-stained endothelial cells clearly separated from surrounding cells. The coefficient of variation (standard deviation) was used to estimate the variability of vessel counts within the same tumor. The mean number of intratumoral microvessels per unit area among different tumors was compared using Student’s *t*-test to identify significant differences.

### 4.9. Statistical Analysis

All experiments were replicated three times in triplicate, and the mean and standard error of the mean were calculated. Statistical analysis was performed to determine the significance of the results. For the in vivo experiment involving a single dose of Leb-C, the treated group was compared to the control group using one-way analysis of variance. Student’s *t*-test was employed as a statistical approach. The significance levels considered acceptable were *p* < 0.05 (*) and *p* < 0.01 (**).

## 5. Conclusions

In conclusion, this study highlights the significant potential of Leb-C, a disintegrin-like protein derived from snake venom, as a novel therapeutic agent for breast cancer treatment. The in vitro and in vivo results demonstrate its remarkable anti-angiogenic activity, leading to inhibition of tumor growth and metastasis. These findings provide valuable insights into the development of anti-tumor and anti-angiogenic strategies to improve TNBC treatments. 

## Figures and Tables

**Figure 1 ijms-24-12219-f001:**
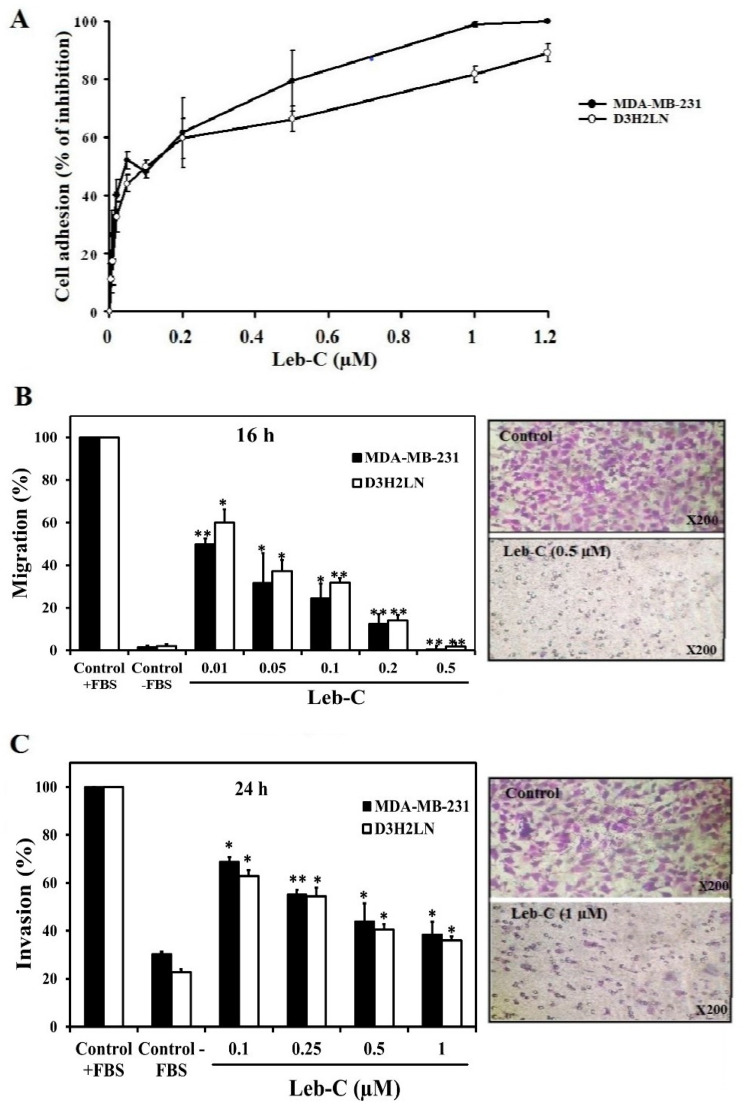
Inhibition of breast cancer cell adhesion, migration, and invasion by Leb-C. (**A**) Adhesion assay of MDA-MB-231 and D3H2LN cells on fibronectin. Cells were pre-incubated with increasing concentrations of Leb-C and seeded onto Fn-coated plates. Adhered cells were stained and quantified. Data represent the mean ± SD of three independent experiments. (**B**,**C**) Migration and invasion assays, respectively, of MDA-MB-231 and D3H2LN cells treated with Leb-C. Cells were pre-incubated with Leb-C and added to Boyden chambers. After incubation, invading cells were stained and counted. The columns represent the mean ± SD of three independent experiments. Representative images show the maximum effect of Leb-C on MDA-MB-231 cell migration (0.5 µM) and invasion (1 µM) compared to the control group. Statistical significance was determined by (*) for *p* < 0.05 and (**) for *p* < 0.01 compared to control.

**Figure 2 ijms-24-12219-f002:**
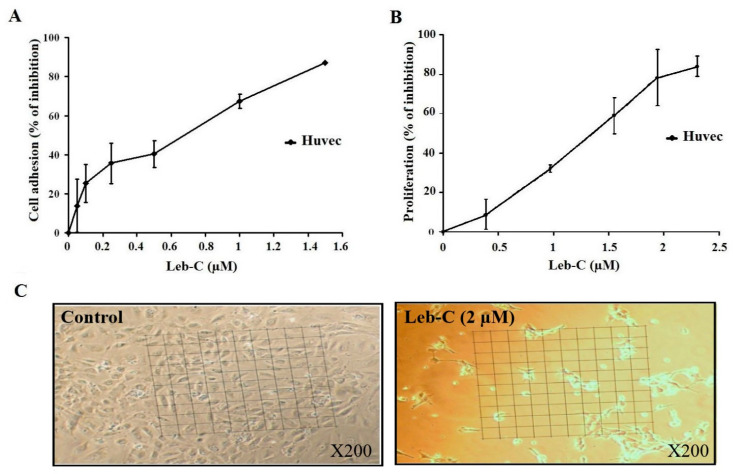
Inhibition of HUVEC cell adhesion and growth by Leb-C. (**A**) Adhesion assay of HUVECs on fibronectin. Cells were pre-incubated with different concentrations of Leb-C and seeded onto Fn-coated plates. Adhered cells were stained, and the absorbance was measured. Data represent the mean ± SD of at least three independent experiments. (**B**,**C**) Viability analysis of HUVECs treated with Leb-C. Viability kinetics of HUVECs treated with increasing concentrations of Leb-C for 72 h. Light microscopy images of HUVECs after 72 h of treatment with Leb-C (2 µM) compared to the control group (magnification 200).

**Figure 3 ijms-24-12219-f003:**
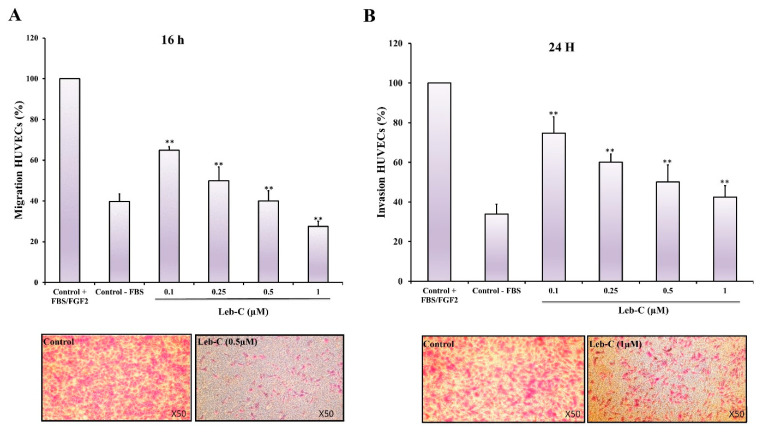
**Effect of Leb-C on HUVEC cell migration and invasion** (**A**) Migration assay of HUVECs treated with Leb-C. Cells were pre-incubated with Leb-C and added to Boyden chambers. After incubation, migrating cells were stained and counted. (**B**) Invasion assay of HUVECs treated with Leb-C. Cells were pre-incubated with Leb-C and added to Matrigel-coated Boyden chambers. After incubation, invading cells were stained and counted. The columns represent the mean ± SD of three independent experiments. Representative images (×50 magnification) show the maximum effect of Leb-C on HUVEC migration (0.5 µM) and invasion (1 µM) compared to the control group. Statistical significance was determined (**) for *p* < 0.01 compared to control.

**Figure 4 ijms-24-12219-f004:**
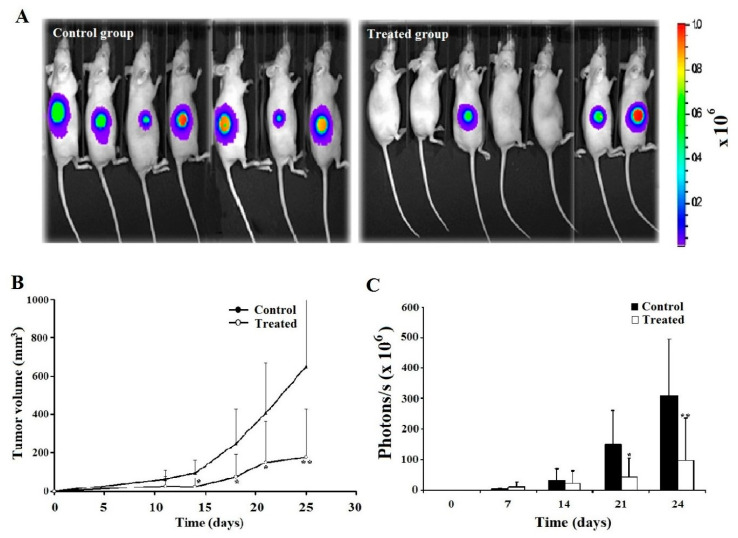
Analysis of D3H2LN xenograft tumor growth inhibition by Leb-C. (**A**) Representative bioluminescence images of control and Leb-C-treated mice. Tumor growth was monitored using the IVIS imaging system. (**B**) Quantification of D3H2LN tumor volume after Leb-C treatment. Tumor volume was measured using calipers, and the data represent the mean ± SD of tumor volume for each group. (**C**) Quantification of bioluminescent signal intensity in Leb-C-treated mice. The data represent the mean ± SD of photon flux per mouse at indicated days. Statistical significance was determined by * *p* < 0.05 and ** *p* < 0.01 compared to the control group.

**Figure 5 ijms-24-12219-f005:**
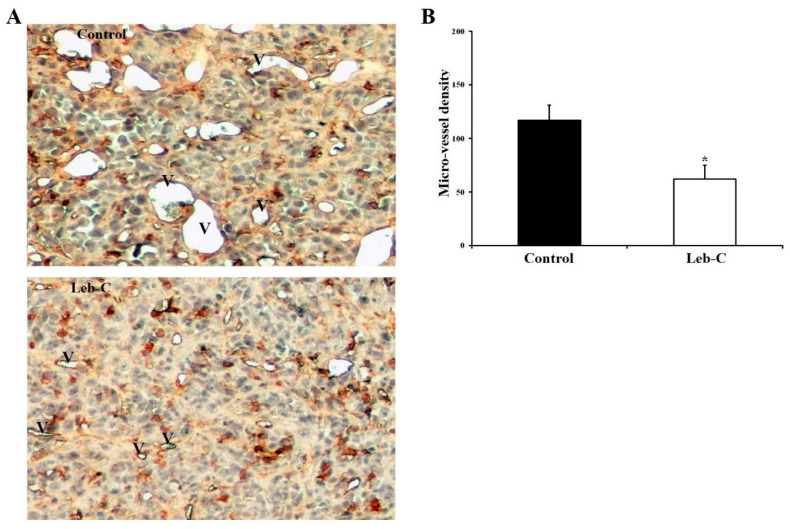
Analysis of intratumor vascular structures in Leb-C-treated tumors. (**A**) Representative images of vessel areas in tumor sections from control and Leb-C-treated groups. Vessel areas (indicated by the letter V) were identified by GSL-1 staining magnification (×250). (**B**) Quantification of intratumor microvessel density. Microvessels were counted and expressed as the highest number per area. Data represent the mean ± SD of microvessel numbers per area in different tumors. Statistical significance was determined by (*) for *p* < 0.05.

## Data Availability

The data presented in this study are openly available in www.researchgate.net/publication/371541805 posted on 13 June 2023 at DOI: 10.20944/preprints202306.0808.v2.

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
