# Peer review of "Disintegrin-like Protein Strategy to Inhibit Aggressive Triple-Negative Breast Cancer"

_ijms, 2023, doi:10.3390/ijms241512219_

Round 1
Reviewer 1 Report
In this article, authors investigate the effects of Leb-C, an active agent isolated from snake venom, on angiogenesis and showed it's antitumor effects in tripple negative breast cancer. The study predominantly lacks any mechanistic insights behind the anti-tumor effects of Leb-C. It is not clear whether integrins and the downstream signaling pathway is affected in this study upon treatment with LebC. How is LebC affecting angiogenesis and tumor growth is not clear from this study. Authors need to conduct experiments to investigate the molecular mechanism of action of LebC.
No additional comments
Author Response
In response to the reviewer's comment, we would like to acknowledge their feedback and emphasize that our study focused on investigating the anti-tumor effects of Leb-C on angiogenesis in triple-negative breast cancer. We agree with the reviewer that further studies elucidating the molecular mechanisms of action are important. However, the data presented in this study highlight Leb-C as a that inhibits resistant several metastatic triple-negative cells. This represents a significant advancement in potential new strategy to overcome chemoresistance as expected by the topic of this special issue.
As a reminder, our previous studies have already demonstrated that Leb-C exerts its effects through an RGD mimetic mechanism involving integrins (please refer to the paper by Limam et al., 2010, doi: 10.1016/j.matbio.2009.09.009). Additionally, we would like to communicate to the referee that new research has evidenced that Leb-C induces apoptosis in cells. We are currently including more extensive signaling data, which are still under investigation, to further elucidate the molecular mechanisms involved.

Reviewer 2 Report
Summary: The authors found that Leb-C (leberagin-C), one essential component from venoms, has a promising effect on TNBC, an aggressive breast cancer type that has no effective treatment. Leb-C is so far the only disintegrin-like protein that possess anti-angiogenesis capability in both in vitro and in vivo. Leb-C can also inhibit adhesion, migration and invasion but not proliferation in MDA-MB-231, D3H2LN and HUVEC, which can synthesize and immobilize Fn for adherence. Mechanistically, leb-C is employing an RGD mimetic mechanism through integrin avb2, avb6 and a5b1.
Overall, this work is largely based on their previous report in 2010 and here, they provide significant biological applications on Leb-C. The manuscript provides clear and sound experiments. The observations are important for the community and therefore suggests for publication:
1. Please label “Figure 1/2/3…etc” on each figure.
2. Previous works from the same lab has tested the effect on Leb-C in different cancer cell lines, I am wondering does Leb-C work on cancer cell models in general or there are some levels of selectivity?
3. I suggest to include a negative control in Figure 1, 2 and 3.
4. In Result section, I suggest to have a brief rationale to describe the cell lines switch from TNBC to HUVEC.
5. Resolutions in Figure 2B and Figure 3 are not very clear.
Author Response
Thank you for your valuable feedback and positive assessment of our work. We have carefully considered your suggestions and have made the following revisions to address your comments:
- We agree and we added the number of each Figure.
- Regarding the selectivity of Leb-C on different cancer cell models, our previous work has extensively investigated its impact on the function of integrin adhesion receptors. We specifically examined the effects of Leb-C on various cancer cell lines in conjunction with different extracellular matrix (ECM) protein pairs, each involving a distinct integrin receptor. Our results demonstrated that Leb-C did not induce any modifications in cell adhesion mediated by alpha1beta1, alpha2beta1, alpha6beta4, or alphavbeta5 integrins. However, it effectively blocked the adhesive function of alphavbeta3, alphavbeta6, and alpha5beta1 integrins. To further elucidate the specificity of Leb-C towards these integrins, we performed additional investigations using IGR39 cells, which are derived from melanoma and overexpress these integrins (please, let’s see our paper doi :10.1016/j.matbio.2009.09.009).
Based on our results, we anticipate that the effects of Leb-C will be highly selective in cancer cells that express alphavbeta3, alphavbeta6, and alpha5beta1 integrins. This information is particularly relevant to our study on triple-negative breast cancer (TNBC) cells, as literature suggests that these cells exhibit overexpression of integrin alphavbeta3 and alpha5beta1 on their surfaces. Similarly, endothelial cells are known to express integrin alphavbeta3.
By examining the effects of Leb-C on these specific cell types, we aimed to shed light on its potential therapeutic applications and explore its ability to target key components involved in tumor growth and angiogenesis. These findings contribute to a deeper understanding of Leb-C's selectivity and its potential as a targeted therapy for cancer.
- We sincerely appreciate your suggestion to include a negative control. In response, we have taken your feedback into consideration and have included appropriate negative control samples in our revised manuscript. These control samples serve as a baseline comparison for the migration and invasion assays presented in Figure 1 and Figure 3.
Concerning Figure 2, only positive control (defined as cells with optimal medium alone) was used since we investigated the inhibitory effect of Leb-c on cell proliferation and adhesion.
We hope that the inclusion of the negative control samples in Figures 1 and 3 addresses your concern and provides a more comprehensive representation of our experimental results.
- We agree. In response to the reviewer's suggestion, we have included a brief rationale to explain the switch from triple-negative breast cancer (TNBC) cell lines to human umbilical vein endothelial cells (HUVEC) in our study.
In the Results section, page 6, line 4:
“To investigate the role of Leb-C on angiogenesis, which are key players in the process tumor progression and treatment response, we used human umbilical vein endothelial cells (HUVECs).”
- We apologize for the lack of clarity in Figure 2B and Figure 3. We have now enhanced the resolution of these figures to improve visibility and ensure better interpretation of the presented data.
Finally, we appreciate your valuable input, and we believe that these revisions have strengthened our manuscript. Thank you for your support and consideration.
